# The Janus Face of p53-Targeting Ubiquitin Ligases

**DOI:** 10.3390/cells9071656

**Published:** 2020-07-09

**Authors:** Qian Hao, Yajie Chen, Xiang Zhou

**Affiliations:** 1Fudan University Shanghai Cancer Center and Institutes of Biomedical Sciences, Fudan University, Shanghai 200032, China; 18111230019@fudan.edu.cn; 2Department of Radiation Oncology, Fudan University Shanghai Cancer Center, Fudan University, Shanghai 200032, China; 16111230008@fudan.edu.cn; 3Key Laboratory of Breast Cancer in Shanghai, Fudan University Shanghai Cancer Center, Fudan University, Shanghai 200032, China; 4Shanghai Key Laboratory of Medical Epigenetics, International Co-laboratory of Medical Epigenetics and Metabolism, Ministry of Science and Technology, Institutes of Biomedical Sciences, Fudan University, Shanghai 200032, China

**Keywords:** p53, mutant p53, E3 ubiquitin ligase, tumor suppressor gene, oncogene, cancer therapy

## Abstract

The tumor suppressor p53 prevents tumorigenesis and cancer progression by maintaining genomic stability and inducing cell growth arrest and apoptosis. Because of the extremely detrimental nature of wild-type p53, cancer cells usually mutate the *TP53* gene in favor of their survival and propagation. Some of the mutant p53 proteins not only lose the wild-type activity, but also acquire oncogenic function, namely “gain-of-function”, to promote cancer development. Growing evidence has revealed that various E3 ubiquitin ligases are able to target both wild-type and mutant p53 for degradation or inactivation, and thus play divergent roles leading to cancer cell survival or death in the context of different p53 status. In this essay, we reviewed the recent progress in our understanding of the p53-targeting E3 ubiquitin ligases, and discussed the potential clinical implications of these E3 ubiquitin ligases in cancer therapy.

## 1. The Wild-Type p53 Signaling Network in Tumor Suppression

The carcinogenesis is a complex event involving mutation or dysregulation of tumor suppressor genes, and amplification or activation of oncogenes. Wild-type p53 is the most influential tumor suppressor in human cancer. In response to a variety of stress signals, p53 is activated to inhibit malignant transformation, tumor cell growth, proliferation and motility, and cancer-fostering metabolism through different mechanisms.

### 1.1. Wild-Type p53 Suppresses Tumorigenesis and Cancer Progression

The tumor suppressor gene *TP53* encodes a transcription factor, p53, consisting of three functional protein motifs, the transactivation domain (TAD), DNA-binding domain (DBD), and tetramerization domain (TD), critical for its tumor-suppressive activity. It has been well documented that p53 harnesses tumor growth and propagation through various mechanisms [1,2]. While pre-cancerous cells undergo genomic instability caused by oncogene activation or carcinogens, p53 is immediately activated to induce the expression of, for example, p21, BTG2, GADD45A, DDB2, and FANCC, consequently leading to cell cycle arrest and DNA repair [3,4]. In addition, p53 can transcriptionally induce the expression of antioxidant genes, such as GPX1, GLS2, and TIGAR, which remove the excessive reactive oxygen species, thereby protecting the genome from the oxidative insult [5]. Thus, p53 is regarded as the guardian of the genome to prevent cells from malignant transformation. Once the pre-cancerous lesion is progressing to the malignancy, p53 is able to drive cancer cell apoptosis by upregulating the expression of multiple apoptotic genes, such as PUMA, BAX, NOXA, and APAF1, in response to different anti-cancer treatments that elicit DNA damage or ribosomal stress [6,7,8,9]. Another irreversible effect of p53 activation on the clearance of cancer cells is to provoke senescence, a permanent cell cycle response, through induction of p21, PAI-1, and PML [10]. In some cases, p53 can eliminate cancer cells through autophagy or autophagic cell death by transcriptionally elevating the expression of genes, such as AMPK, DRAM, and SESN2 [11]. In addition, p53 can also evoke ferroptosis through transcriptional repression of SLC7A11, a key component of the cystine/glutamate antiporter [12]. Recently, it has been found that the complex role of p53 in fine-tuning cancer cell survival and death is also reflected by its activity in maintaining metabolic health and disturbing cancer-favouring metabolism [13,14] by orchestrating the biosynthesis or homeostasis of glucose [15,16], cholesterol and nonsterol isoprenoids [17], cardiolipin [18], polyamine [19], α-ketoglutarate [20], and essential amino acids [21,22]. Altogether, p53 can suppress tumor initiation and progression through multiple mechanisms through transcriptional regulation.

Besides acting as a transcription factor, the cytoplasmic p53 has also been found to play roles in tumor suppression [23]. The cytosolic p53 can directly derepress the mitochondrial BAK, BAX and PUMA by interacting with Bcl-2 or Bcl-XL, consequently leading to mitochondrial outer membrane permeabilization (MOMP) and the release of cytochrome c [24,25]. A later study described that PUMA releases p53 from Bcl-XL binding and thus enhances p53-mediated mitochondria apoptotic pathway, suggesting a dynamic interplay between p53 and the Bcl-2 family proteins in the modulation of apoptosis [26]. Interestingly, it was found that depletion or inactivation of p53 induces autophagy in several cell lines and across species, which improves cancer cell survival under hypoxia and nutrient-restricted conditions [27]. The study further demonstrated that the cytoplasmic, but not the nuclear, p53 is able to attenuate the enhanced autophagy upon adverse situations [27]. Thus, the cytosolic p53 also executes tumor-suppressive function via transcription-independent mechanisms.

### 1.2. Regulation of Wild-Type p53 Activity

As the cytotoxic effect of p53 is so detrimental to cancer cells, a plethora of mechanisms have been evolved to inhibit p53 activity in order for tumors to survive and propagate (mutation of p53 occurs in the rest of the tumors, which is discussed in a later section). For instance, the E3 ubiquitin ligase MDM2 and its homologous partner, MDMX, which are often amplified or overexpressed in cancer [28,29], have been demonstrated as the master antagonists against p53. MDM2 is encoded by a p53-inducible gene and dampens p53 activity through multiple mechanisms, thus forming a negative feedback circuit. First, MDM2 induces poly-ubiquitination and proteolytic degradation of p53 by interacting with the latter [30,31,32]. Second, MDM2 also mediates mono-ubiquitination of p53, leading to translocation of p53 to the cytoplasm and abrogation of its transcriptional activity [33]. Third, MDM2 prevents p53 communication with its target gene promoters and thus represses p53-target gene expression by directly associating with the TAD of p53 [34,35,36,37]. Finally, MDM2 suppresses p53 mRNA translation by prompting proteasomal degradation of the ribosomal protein RPL26 and dissociating the RPL26–p53 mRNA interaction [38]. In most cases, MDM2 was shown to work with MDMX to proficiently hamper p53’s function [39]. Although MDMX lacks E3 ligase activity, it associates with MDM2 through their C-terminal RING-finger domains and boosts the E3 ligase activity of the latter toward p53 [40,41]. The central role of MDM2 and MDMX in controlling p53 activity is also gracefully approved by in vivo genetic studies, as the early embryonic lethality caused by knocking out MDM2 or MDMX is completely restored by simultaneously deleting p53 [42,43,44]. Nevertheless, MDM2- or MDMX-mediated inhibition of p53 can be overcome by various stress signals, such as DNA damage and nucleolar stress, as detailed below. Thus, in support of cancer cell growth, additional E3 ubiquitin ligases and oncogenic molecules have evolved to directly inactivate p53 or potentiate MDM2’s E3 activity toward p53. Recently, we have demonstrated that NGFR and PHLDB3, which are overexpressed in melanoma, neuroblastoma, colorectal cancer and lung cancer, can bind to MDM2 and/or p53, leading to forced inactivation of p53 and chemoresistance of cancer cells [45,46]. Most importantly, more than a dozen molecules, including MDM2, NGFR, and PHLDB3, were found to be transcriptionally induced by p53, thus constituting the auto-regulatory network confining p53 activity [47]. Owing to the close surveillance by these negative modulators, p53 protein maintains a relatively low level in cancer cells.

However, it has been found that p53 can be modulated and activated via diverse post-translational modifications (PTMs) in response to various stress signals, including DNA damage, nucleolar stress, hypoxia, oxidative stress, telomere erosion, and oncogene activation [1,2,48,49]. For instance, the DNA damage insult mediates phosphorylation of p53 at Ser15 and Ser20 by ATM, ATR, DNA-PK, Chk1, and Chk2, resulting in an increase of transcriptional activity of p53 [50,51,52]. DNA damage-mediated acetylation of p53 at Lys370, Lys372, Lys373, Lys381, Lys382, and Lys386 by CBP/p300 may prevent MDM2-induced p53 ubiquitination and degradation, as ubiquitination and acetylation are mutually exclusive modifications [53]. Interestingly, acetylation of specific lysines may affect p53 preference for binding to distinct promoters of its target genes. Lys120, located within the DBD, is acetylated upon DNA damage stress, which is required for induction of proapoptotic genes, such as PUMA and BAX [54,55]. Simultaneous loss of acetylation of the above mentioned seven lysines plus Lys146 can completely abolish p53 ability to induce cell cycle and apoptotic genes [56], indicating that acetylation is indispensable for p53 transcriptional activity.

Over the past decade, the nucleolus has emerged as a considerable cellular compartment monitoring the p53 signaling pathway [7,57]. The first interaction between the ribosomal protein RPL5 and MDM2 was described in 1994 [58]. However, whether this interaction is relevant to regulation of the p53 pathway and cancer development had remained elusive for almost 10 years until RPL11, RPL5, and RPL23 were reported to regulate the MDM2-p53 feedback loop under nucleolar stress (also known as ribosomal stress) [7,59]. Ribosome biogenesis, majorly taking place in the nucleolus, is a tightly organized multistep process involving four ribosomal RNA (rRNA) species and 79 ribosomal proteins, as well as >150 ribosome biogenesis-related factors. Perturbation of any steps of this process triggers nucleolar stress, leading to the accumulation of ribosome-free ribosomal proteins in the nucleoplasm. These dissociative ribosomal proteins can interact with MDM2 or p53 and block MDM2-induced p53 ubiquitination and proteolytic degradation [7,59,60]. Apart from the ribosomal proteins, several ribosome biogenesis-related proteins have also been found to stabilize p53 by interplaying with the MDM2-p53 circuit in response to nucleolar stress. We recently identified the ribosome maturation protein SBDS as a binding partner of p53, though it also slightly interacts with MDM2 [61]. SBDS associates with p53 at the N-terminal TAD that is also the region for MDM2 binding. The interaction of SBDS and p53 impairs formation of the MDM2-p53 complex resulting in p53 activation upon nucleolar stress [61]. Thus, these nucleolar proteins serve as a group of activators of p53 by counteracting MDM2’s function. Importantly, interference with the nucleolar integrity has been employed in the development of anti-cancer approaches, leading to the discovery of small molecules, CX-3543 and CX-5461, which selectively suppress tumor growth through the nucleolar stress-p53 signaling pathway by inhibiting RNA Pol I-dependent transcription [62,63].

Taken together, p53 can execute its tumor-suppressive function through both transcription-dependent and -independent manners. Although p53 usually maintains a restricted level due to overexpression of a great many oncogenic inhibitors in cancer cells, it can be activated in response to various stress signals, which is critical for the development of strategies for cancer treatment.

## 2. p53 Mutation Endorses Cancer Development

*TP53* is the most frequently mutated gene in human cancers. For instance, it is mutated in over 95% of high-grade serous ovarian cancer [64,65], 74% of triple-negative breast cancer [66], 50–80% of lung cancer, and 60–70% of pancreatic cancer [67]. The overall frequency of *TP53* mutation across all cancer types is around 50% [67]. The cancer-associated p53 mutants include missense, frameshift, truncation, and deletion mutations, most of which are missense mutations (~74%), a single amino acid substitution that usually occurs in the p53 DBD, including the hotspot mutants, such as R175H, G245S, R248Q, R248W, R249S, R273H, R273C, and R282W. Generally, p53 mutants fall into two distinct categories according to their biochemical features. DNA-contact mutants replace the amino acids critical for DNA binding, such as R248Q, R273H, and R282W, while conformational mutants produce unfolded structure or altered conformation, such as R175H, G245S, and R249S [68].

The emergence of missense mutations can abrogate the tumor-suppressive function of wild-type p53 (wtp53), because the mutations in the DBD prevent mutant p53 (mtp53) from binding to the p53-responsive DNA elements on the target gene promoter. In addition, mutation of one allele of *TP53* can repress the activity of the remaining wild-type allele in cancer cells. Two possible mechanisms have been described to account for this ‘dominant-negative’ effect. First, mtp53 may form cotetramers with its wild-type counterpart leading to the abolishment of its transcriptional activity [69]. Moreover, mtp53 promotes misfolding and coaggregation of wtp53 into cellular inclusions, which insulates the latter from the DNA elements [70]. Remarkably, a number of missense mutants, including the hotspot mutants and several others, such as Y220C, S241F, and R280K, have been demonstrated to drive tumor growth and metastasis by exerting oncogenic function, the so called “gain-of-function” (GOF), independently of wtp53 [71]. p53 mutants have been shown to implement the GOF by interacting with other transcription factors or co-factors. For instance, the p53 mutants, R175H and R280K, were found to prompt TGF-β-induced cancer cell metastasis by boosting Smad2/3 activity and inhibiting the metastasis suppressor, TAp63 [72,73]. mtp53-R273H bolsters the mevalonate pathway by interacting with the sterol regulatory element binding transcription factors (SREBFs), leading to enhanced biosynthesis of sterols and isoprenoids and thus cancer progression [74]. It has been recently described that a liver cancer-derived hotspot p53 mutant, R249S, can be phosphorylated by CDK4/Cyclin D and translocate into the nucleus, where it empowers the c-Myc-induced ribosome biogenesis and hepatocellular carcinoma cell proliferation [75,76]. Like wtp53, mtp53 also performs its oncogenic function independent of transcriptional regulation. For instance, p53 mutants were found to hamper the activation of caspase-8 [77] and caspase-9 [78], and inhibit the cleavage of caspase-3 [79] through directly binding to these proteins, leading to reduced apoptosis and cancer cell survival and proliferation. In other scenarios, the hotspot p53 mutants, R248W and R273H, could interact with the MRN (Mre11-Rad50-NBS1) complex that is required for DNA double-stranded break repair. This interaction leads to induction of replication stress and impairment of DNA damage response, resulting in the progression of cancer [80]. The mtp53 GOF was also elegantly validated by the genetic knock-in mice bearing mtp53-R172H (equivalent to human R175H) or R270H (equivalent to human R273H), exhibiting augmented cell proliferation, DNA synthesis, transformation potential and more aggressive tumor phenotypes compared with the p53-null mice [81,82].

Although the vast majority of p53 mutants lose the wild-type function or exert a ‘dominant-negative’ effect on the remaining wild-type allele in cancer cells, many of them, particularly the missense and hotspot mutations, promote cancer development by acquiring the oncogenic GOF, and are therefore potential targets for anti-cancer therapy.

## 3. Divergent Roles of p53-Targeting E3 Ligases in Cancer

Although missense mutations mostly occur in the DBD and may locally or conformationally affect the structure and function of the p53 proteins, these mutants have been shown to share a multitude of binding partners with wtp53. Accumulating evidence has revealed that numerous E3 ubiquitin ligases can target both wt and mtp53 for degradation or inactivation (Table 1). It is thus reasonable to postulate that these E3 ubiquitin ligases may possess oncogenic function in wtp53-harboring cancer cells, whereas they play a tumor-suppressive role under the circumstance of p53 mutation. By taking this into consideration, the therapeutic approaches targeting these E3 ubiquitin ligases or their interactions with p53 should be re-assessed with regard to p53 status in cancer.

### 3.1. RING-Finger E3 Ligases

The RING-finger E3 ligase MDM2, encoded by a wtp53 target gene, interacts with and induces ubiquitination and proteolytic degradation of wtp53, as thus forming a negative feedback circuit, which plays a central role in the surveillance of wtp53 stability and activity [49,52]. Consistently, the MDM2-encoding gene was found to be overexpressed or amplified in various human cancers [28]. Therefore, efforts had been made for decades to develop small molecules to block MDM2 E3 ligase activity or disrupt MDM2-p53 association, leading to the discovery of Nutlin-3a and its derivatives [118,119]. This group of compounds interact with MDM2 in the p53-binding pocket, which disrupts the association of MDM2 with wtp53 and enhances wtp53 stability and activity. The Nutlin-3a family of MDM2 antagonists displays anti-cancer activity by selectively inducing wtp53-dependent cell cycle arrest, apoptosis and in vivo tumor growth inhibition (Figure 1). Interestingly, it has been found that MDM2 is also responsible for mtp53 degradation in vivo and in tumors [83], which suggests that MDM2 may serve as a tumor suppressor in mtp53-sustaining cancer. By this token, the MDM2-targeting drugs, such as Nutlin-3a, could also stabilize mtp53, leading to a tumorigenic consequence. Indeed, several studies have revealed that Nutlin-3a increases the mtp53 level by perturbing the MDM2–mtp53 interaction in zebrafish and in cancer cells [93,120]. In fact, the strategy bolstering MDM2-mediated mtp53 degradation has been employed in treating tumors harboring mutant p53, which is discussed in a later section. Therefore, strategies targeting MDM2 should be cautiously employed, considering that MDM2 plays both oncogenic and tumor-suppressive roles via degrading wt and mtp53, respectively.

*RNF128* encodes two isoforms of the ring finger protein RNF128 that was characterized as a crucial regulator of T cell function, cytokine production, and nutrient metabolism through in vitro and in vivo studies [121,122,123]. A recent study demonstrated that RNF128, namely Grail, which is also transcriptionally induced by wtp53, can in turn physically and functionally interact with the N-terminus of wtp53 to induce its proteasomal degradation, resulting in the inhibition of cancer cell apoptosis [86]. Interestingly, RNF128 seemed to modulate p53 transcriptional activity and determine cell fate by differentially regulating cell cycle and apoptotic gene expression, as loss of RNF128 expression induced p21-dependent cell cycle arrest, but not p53-dependent apoptosis [86], revealing an unique regulatory mode for controlling p53 activity. On the contrary, it has also been reported that the E3 activity-proficient isoform of RNF128 can behave as a tumor suppressor by efficiently ubiquitinating and degrading mtp53 and prohibiting progression of esophageal adenocarcinoma [87]. The other isoform of RNF128, with limited E3 activity, could negatively regulate mtp53 ubiquitination by forming heterodimers with the former isoform. Importantly, simvastatin, a mevalonate inhibitor widely used for treatment of cancer, significantly diminished the level of the E3 activity-limited isoform of RNF128, thus triggering degradation of mtp53 by the E3 activity-proficient RNF128 (Figure 1) [87]. Together, these studies have elaborated that the E3 ligase activity of RNF128 is responsible for the degradation of both wt and mtp53, and have suggested that the use of statin may achieve optimum benefit in mtp53-expressing tumors.

Two more members of the RING-finger E3 ubiquitin ligase family, PIRH2 and COP1 that are also encoded by wtp53 target genes, were found to negatively regulate wtp53 stability by promoting its proteolytic degradation [47,88,90]. The RING-finger domain of PIRH2 is required for degradation of wtp53, but dispensable for PIRH2-mediated inhibition of wtp53, as RING-deleted PIRH2 can still suppress wtp53 transcriptional activity by interfering with the DNA-binding capacity of wtp53 [88]. Interestingly, Cop1-deficient mice did not show any increase in p53 level and activity [124]; it is therefore likely that this E3 ubiquitin ligase might only be utilized by cancer cells to constrain wtp53 activity. These findings suggest that PIRH2 and COP1 are tumor-promoting proteins by inactivating wtp53. Recent studies also showed that the two E3 ubiquitin ligases are able to target mtp53 for degradation as well [89,91]. An anionic cell-penetrating peptide, p28, with anti-tumor activity was found to stabilize both wt and mtp53 by impeding COP1 and p53 interactions (Figure 1) [91]. Arsenic trioxide, a drug for acute promyelocytic leukemia, was shown to promote mtp53 degradation via elevating the expression of PIRH2 (Figure 1) [89]. Additional studies further proposed that both E3 ligases might act as tumor suppressor proteins. It was found that PIRH2 interacts with and promotes proteolysis of c-MYC in cancer cells. Consistently, Pirh2-depleted mice displayed elevated expression of c-Myc and were predisposed to developing tumors. PIRH2 was also found to be underepxressed in lung, ovarian, and breast cancers, which was correlated with unfavorable prognosis of patients [125]. COP1 was found to play a tumor-suppressive role by degrading the oncogenic ETS transcription factors, and loss of COP1 promoted cell proliferation, hyperplasia, and early prostate intraepithelial neoplasia [126]. Thus, more efforts are necessary to elucidate the roles of PIRH2 and COP1 in the context of different cancers and p53 genotypes, which would be informative for the development of cancer therapies by targeting the two RING-finger E3 ligases.

### 3.2. Tripartite Motif Protein Family

The tripartite motif protein TRIM71 that also possesses a RING-finger motif with E3 ligase activity was found to interact with and degrade wtp53 through the ubiquitin-proteasome system leading to restricted apoptosis during neural stem cell differentiation and early brain development [92]. It had been elusive if TRIM71 plays a role in cancer by regulating the p53 pathway, until we recently uncovered the interplay between TRIM71 and wt/mtp53 in ovarian cancer [93]. Through the co-immunoprecipitation assay coupled with mass spectrometry, we identified TRIM71 as an interacting protein of mtp53 in ovarian cancer tissues. This interaction prompted the ubiquitination and degradation of p53 mutants, leading to the inhibition of ovarian cancer cell growth and invasion. Consistently, high levels of TRIM71 in ovarian carcinomas predicts favorable prognosis of ovarian cancer patients. Additionally, we also found that Nutlin-3a, which disrupts the MDM2-mtp53 complex, enhances the interaction of TRIM71 with mtp53, while the HSP90 inhibitor, which promotes mtp53 degradation by MDM2, reduces TRIM71 binding to mtp53, revealing a competitive p53-binding pattern of MDM2 and TRIM71. Thus, our finding has demonstrated that TRIM71 and MDM2 act as tumor suppressors by competitively degrading mtp53 in ovarian cancer [93]. Together with the other study showing degradation of wtp53 by TRIM71 in the brain or nervous system [92], it is believed that TRIM71 could be able to dampen both wt and mtp53 activity relying on different tissues or cancer types.

Another tripartite motif protein, TRIM24, was initially described as a liver-specific tumor suppressor. In Trim24-null mice, hepatocytes were prone to undergo malignant transformation and progress to metastatic hepatocellular carcinoma by upregulating the retinoic acid signaling [127]. However, it was later found that this protein can serve as an oncogenic co-activator of estrogen receptor (ER) [128], androgen receptor (AR) [129] and the STAT3 signaling pathway [130] in breast, prostate, and brain cancers, respectively. In line with these studies, TRIM24 was identified as an evolutionarily conserved negative regulator of wtp53, by promoting wtp53 ubiquitination and destruction via its RING-finger domain, from Drosophila to mammalian cells, and was suggested as a therapeutic target to restore p53 activity in cancer. [94]. Intriguingly, it was reported that TRIM24 could orchestrate mtp53 function through a distinct mechanism independent of its E3 ligase activity. TRIM24 displayed a tumor inhibitory effect and protected mouse embryonic stem cells from malignant transformation by converting endogenously expressed mtp53-R172H to a wild-type conformation [95]. Thus, the p53 status and cancer types should be taken into account if targeting TRIM24 is considered as a therapeutic approach.

An additional case is the tumor suppressor TRIM19, better known as PML, that was found to induce wtp53 stabilization and activation through multiple mechanisms. Although having a RING-finger motif, TRIM19 has not been reported to possess E3 ligase activity. Instead, it enhances wtp53 acetylation through CBP [96] and MOZ [100], and phosphorylation by HIPK2 [97] and CK1 [99] in response to oncogenic or DNA damage stress, consequently protecting wtp53 from MDM2-mediated ubiquitination and degradation [98]. Although multiple studies pointed out that TRIM19 acts as a tumor suppressor by potentiating wtp53 activity, this TRIM protein also displays the other face by regulating mtp53. In mtp53-expressing colon and breast cancer cells, TRIM19 binds to and enhances transcriptional activity of mtp53, resulting in accelerated cell proliferation [101]. Altogether, at least three TRIM proteins have been validated to gain both oncogenic and tumor-suppressive functions via the regulation of wt and mtp53.

The tripartite motif protein family comprises over 70 members, most of which contain a RING-finger motif displaying E3 ubiquitin ligase activity. Besides the above three TRIM proteins, a multitude of family members have been shown to modulate wtp53 activity [131]. For instance, TRIM32 [103], TRIM39 [104], TRIM59 [105], and TRIM66 [106] are able to promote polyubiquitination and degradation of wtp53. TRIM21 [107], TRIM25 [108] and TRIM28 [109] trigger wtp53 destabilization by facilitating MDM2-mediated p53 ubiquitination. A RING-lacking TRIM protein, TRIM29, can relocate wtp53 from the nucleus to the cytoplasm leading to impairment of wtp53 transcriptional activity [110]. In addition, several TRIM proteins, such as TRIM8 [111] and TRIM13 [112], can stabilize and activate wtp53 by degrading MDM2 or inhibiting its E3 ligase activity toward wtp53. However, it remains unclear whether these TRIM proteins also regulate p53 mutants. Further efforts are needed for trimming the potential dual activity of these TRIM proteins in cancers harboring wt or mtp53.

### 3.3. Others

The U-box E3 ligase CHIP (C-terminus of HSP70-interacting protein) was reported to induce degradation of wt and mtp53 involving UbcH5b as an E2 and the chaperone protein HSP70 as a bridge between the E3 ligase and p53s [113]. CHIP appears to reduce the level of mtp53-R175H more markedly than that of wtp53, because HSP70 often chaperones unfolded proteins, like the conformational mutant R175H [132]. By associating with another chaperone HSP90, p53 mutants are usually stabilized and maintained at a high level in cancer [84], which leads to the strategy of targeting HSP90 for the treatment of cancers sustaining mtp53. For instance, the HSP90 inhibitor tanespimycin (17-AAG) can destabilize p53 mutants, such as p53-V143A, R175H, S241F, R273C/H, and R280K, by inactivating HSP90 and thus boosting MDM2- or CHIP-mediated mtp53 degradation (Figure 1) [84,85]. Additionally, an HDAC inhibitor suberoylanilide hydroxamic acid (SAHA) was also shown to preferentially suppress cancer cells with p53 mutations by inhibiting HDAC6, an essential positive regulator of HSP90 (Figure 1) [133]. Thus, targeting the CHIP–mtp53 cascade could be a feasible strategy for treating mtp53-expressing carcinomas.

In our recent work, by screening mtp53-interacting partners in ovarian cancer tissues harboring mtp53 [93], several wtp53-targeting E3 ubiquitin ligases were identified as mtp53-associated partners, including Cullin 4B [114], Cullin 7 [115], Cullin 9 [116], and HUWE1 [117]. These E3 ubiquitin ligases were shown to bolster cancer cell survival and proliferation by inhibiting wtp53 activity. Mechanistically, the Cullin family E3 ligases, Cullin 4B and Cullin 7, and the HECT motif-containing E3 ligase, HUWE1, are able to mediate mono- or poly-ubiquitination of wtp53, while Cullin 9, also regarded as a Parkin-like E3 ligase, can sequester wtp53 in the cytoplasm independently of its E3 ligase activity [116]. In the light of our finding [93], it is postulated that these E3 ligases may also play a tumor-suppressive role by opposing mtp53 in ovarian cancer, which is tempting to investigate in a future study.

## 4. Concluding Remarks

Although it is now evident that p53 is an essential determinant for the role of its associated E3 ubiquitin ligases, they do show some preference to regulate different cancer-related signaling pathways in certain cancer types. For example, TRIM24 was able to boost the ER, AR, and STAT3 pathways in breast, prostate and brain cancers, respectively [128,129,130], in addition to targeting wtp53 for degradation in breast cancer [94]. Given that all the three types of cancer sustain a low frequency of p53 mutation [67], these findings strongly suggest that TRIM24 could be a tumor promoter in wtp53-harboring tumors. However, it is not clear whether TRIM24 induces the ER, AR, and STAT3 pathways in p53-highly mutated cancers, such as epithelial ovarian cancer, and, if so, what is the consequence if TRIM24 activates those oncogenic signaling pathways on one hand and suppresses mtp53 on the other? We recently reported that inhibition of mtp53 by enhancing MDM2 E3 ligase activity may not be the optimal strategy [93], as MDM2 can serve as a p53-independent oncogene [134,135] that is amplified in multiple human cancers [28]. Under this scenario, potential compounds selectively disrupting the interaction of the E3 ligase with wt or mtp53, rather than merely targeting the E3 ligase itself, should be more precisely and effectively applicable in the treatment of cancer. Hence, it is also worthwhile and necessary to elucidate the role of the p53-targeting E3 ubiquitin ligases in the regulation of other cancer-related signaling pathways in different cancer types, which would be beneficial to the optimum use of targeting therapies.

Interestingly, it is noticed that many of the E3 ubiquitin ligases, including MDM2, RNF128, PIRH2, COP1, TRIM19 and TRIM24, are encoded by wtp53-inducible genes. This may partially explain why these E3 ligases are overexpressed in wtp53-harboring cancer as tumor promoters (genomic amplification should be another reason), whereas underexpressed in mtp53-harboring cancer as tumor suppressors. Chemotherapy or radiotherapy-induced DNA damage stress has been shown to induce wtp53 activation and, thus, the expression of these E3 ubiquitin ligases as negative feedback regulators for wtp53. Under such circumstances, it is therefore rewarding to investigate if targeting these E3 ligases can untie the auto-regulatory loops and potentiate the efficacy of the traditional cancer therapies.

In conclusion, the p53 signaling pathway constitutes an overwhelming intricate regulatory network, and the cancer-associated mutations, particularly the GOF mutants, make it even more complex. Although the missense mutations cause the local or conformational structural alterations to generate a variety of p53 mutants, they do share some common binding partners with wtp53. The diverse functions of the E3 ubiquitin ligases, by targeting both wtp53 and mtp53, not only achieve distinct cellular outcomes leading to cancer cell survival or death, but also open new opportunities for potential targeting therapy. To thoroughly understand the underlying mechanisms is certainly instructive for precision medicine for cancer.

## Figures and Tables

**Figure 1 cells-09-01656-f001:**
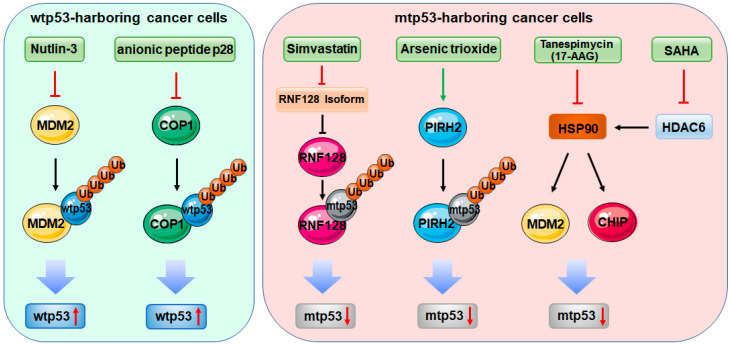
Therapeutic approaches targeting E3 ligases in the context of p53. In wild-type p53 (wtp53)-harboring cancer cells, Nutlin-3 and its derivatives and the anionic peptide p28 stabilize and activate wtp53 by antagonizing MDM2 and COP1, respectively. In mutant p53 (mtp53)-harboring cancer cells, Simvastatin, arsenic trioxide, tanespimycin (17-AAG), and suberoylanilide hydroxamic acid (SAHA) induce mtp53 degradation by directly or indirectly targeting E3 ligases.

**Table 1 cells-09-01656-t001:** Wild-type/mutant p53-targeting E3 ligases in cancer.

E3 Ligases	Mechanism	Reference
RING-finger E3 ligases		
MDM2	Induces ubiquitination and degradation of wtp53; suppresses wtp53 transcriptional activity	[30,31,34,35,36,37]
	Promotes mtp53 degradation	[83,84,85]
RNF128	Induces ubiquitination and degradation of wtp53	[86]
	Promotes mtp53 degradation	[87]
PIRH2	Induces ubiquitination and degradation of wtp53	[88]
	Promotes mtp53 degradation	[89]
COP1	Induces ubiquitination and degradation of wtp53	[90]
	Promotes mtp53 degradation	[91]
*TRIM protein family*		
TRIM71	Induces ubiquitination and degradation of wtp53	[92]
	Promotes mtp53 ubiquitination and degradation	[93]
TRIM24	Induces ubiquitination and degradation of wtp53	[94]
	Converts mtp53-R172H to a wild-type conformation	[95]
TRIM19	Enhances wtp53 acetylation and phosphorylation, and protects it from MDM2-mediated degradation	[96,97,98,99,100]
	Binds to and enhances transcriptional activity of mtp53	[101]
TRIM32, TRIM39, TRIM59, TRIM66	Induces ubiquitination and degradation of wtp53	[102,103,104,105,106]
TRIM21, TRIM25, TRIM28	Promotes MDM2-mediated p53 ubiquitination and degradation	[107,108,109]
TRIM29	Relocates wtp53 from the nucleus to the cytoplasm	[110]
TRIM8, TRIM13	Stabilizes and activates wtp53 by degrading MDM2	[111,112]
*Others*		
CHIP	Induces degradation of both wt and mtp53	[84,85,113]
Cullin 4B	Mediates poly-ubiquitination and degradation of wtp53	[114]
Cullin 7	Mediates mono- or di-ubiquitination of wtp53	[115]
Cullin 9	Sequesters wtp53 in the cytoplasm	[116]
HUWE1	Mediates mono- or poly-ubiquitination of wtp53	[117]

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
