# Peer review of "The Janus Face of p53-Targeting Ubiquitin Ligases"

_cells, 2020, doi:10.3390/cells9071656_

Round 1
Reviewer 1 Report
This is an excellent review for p53 protein stability, although the authors used first part on p53 function as tumor suppressor. The authors should be more clear about p53 role in tumor suppression by cell cycle arrest, apoptosis, ferroptosis etc since the author seems to believe p53 drives apoptosis once pre-cancerous lesion in progressing to the malignancy (line 45). It is widely known p53 can induce all kinds of responses even without transformation, it may be context and stress type dependent. Furthermore, the authors listed all the cytosolic p53 function, however the p53 protein mainly locates in nuclei, and not much in vivo data support this observation. Overall the authors may want to include some in vivo data to support or argue about some of the conclusions. for example, the authors can emphasize mdm2 and mdmx as oncogenes to drive tumorigenesis in vivo by citing mdm2 and mdmx trangene mouse models.
minor points:
line 146, mutant p53 proteins have two groups, contact and structural
line 156, gain of function mouse models are first demonstrated by R172H and R270H, it should be cited .
Table 1, it will be better to include published mouse models to explain the in vivo roles of E3 ligase.
Author Response
Dear Reviewer,
We greatly appreciate your help in improving our manuscript. Listed below are our point-by-point responses.
Overall the authors may want to include some in vivo data to support or argue about some of the conclusions. for example, the authors can emphasize mdm2 and mdmx as oncogenes to drive tumorigenesis in vivo by citing mdm2 and mdmx trangene mouse models.
Response: We appreciate this kind comment. The in vivo studies have been cited and discussed in line 87, “The central role of MDM2 and MDMX in controlling p53 activity is also gracefully approved by in vivo genetic studies, as the early embryonic lethality caused by knocking out Mdm2 or Mdmx is completely restored by simultaneously deleting p53”.
line 146, mutant p53 proteins have two groups, contact and structural.
Response: We are grateful for this suggestion and have modified the contents in line 149, “Generally, p53 mutants fall into two distinct categories according to their biochemical features. DNA-contact mutants replace the amino acids critical for DNA binding, such as R248Q, R273H, and R282W, while conformational mutants produce unfolded structure or altered conformation, such as R175H, G245S, and R249S”.
line 156, gain of function mouse models are first demonstrated by R172H and R270H, it should be cited.
Response: The knock-in mouse models have been cited and discussed in line 180, “The mtp53 GOF was also elegantly validated by the genetic knock-in mice bearing mtp53-R172H (equivalent to human R175H) or R270H (equivalent to human R273H), exhibiting augmented cell proliferation, DNA synthesis, transformation potential and more aggressive tumor phenotypes compared with the p53-null mice”.
Table 1, it will be better to include published mouse models to explain the in vivo roles of E3 ligase.
Response: We agree that the functions of E3 ligases in vivo can be explicitly demonstrated by mouse models. However, the role of the E3s in regulating p53 was not investigated in these mouse models and therefore they are not appropriate to be included in Table 1. Instead, we have described and cited the in vivo studies of Rnf128 in line 224, Cop1 in line 247, Pirh2 in line 257, Trim71 in line 266, and Trim24 in line 284.
Reviewer 2 Report
The review by Hao and its collaborators about p53-targeting ubiquitin ligases is a nice piece of work that will be interested particularly in the p53/ Mdm2 fields.
Though was not extensive, the addition of the impact of E3 ubiquitin ligases on mutant p53 is very welcome in the review.
A few minor suggestions:
It will be important to discuss briefly ( maybe in the concluding remarks section), about the clear redundancy that exists for targeting and degradation of p53 by the ubiquitin ligases and other proteins described in the review. Mdm2, is the main regulator of p53 and in most of the cases seems that it can deal by itself to keep p53 under control. So why there are more of dozen proteins doing the same? What are the biological implications of that and importantly how this diversity plays on mutant p53, cancer and therapeutics?
I found the paragraph describing the two isoforms of RNF128 a little bit confusing ( lines 217 to 233). Maybe describing in the beginning that there are two isoforms of this gene will help.
Finally for the length of the manuscript, 150 references seems are too much.
Author Response
Dear Reviewer,
We greatly appreciate your help in improving our manuscript. Listed below are our point-by-point responses.
It will be important to discuss briefly (maybe in the concluding remarks section), about the clear redundancy that exists for targeting and degradation of p53 by the ubiquitin ligases and other proteins described in the review. Mdm2 is the main regulator of p53 and in most of the cases seems that it can deal by itself to keep p53 under control. So why are there dozens of proteins doing the same? What are the biological implications of that and importantly how this diversity plays on mutant p53, cancer and therapeutics?
Response: We appreciate this insightful comment. Discussion has been made in line 90, “Nevertheless, MDM2- or MDMX-mediated inhibition of p53 can be overcome by various stress signals, such as DNA damage and nucleolar stress, as detailed below. Thus, in support of cancer cell growth, additional E3 ubiquitin ligases and oncogenic molecules have evolved to directly inactivate p53 or potentiate MDM2’s E3 activity toward p53”, and in line 374, “The diverse functions of the E3 ubiquitin ligases by targeting both wtp53 and mtp53 not only achieve distinct cellular outcomes leading to cancer cell survival or death, but also open new opportunities for potential targeting therapy”.
I found the paragraph describing the two isoforms of RNF128 a little bit confusing (lines 217 to 233). Maybe describing in the beginning that there are two isoforms of this gene will help.
Response: We are sorry for the confusion and have made modification in line 224 as suggested.
Finally for the length of the manuscript, 150 references seem too much.
Response: We have deleted several redundant references.
Reviewer 3 Report
In the article entitled: “The Janus face of p53-targeting ubiquitin ligases” by Qian Hao, Yajie Chen and Xiang Zhou, the authors describe nicely the role of ubiquitin ligases in regulation of tumor suppressor p53, both wild type and mutant. The roles of p53 in tumor suppression are extensively described and nice overview of p53-targeting ubiquitin ligases is given.
The only concern is that the authors haven’t mentioned that p53 has multiple isoforms that are expressed in tumors, and their regulation by ubiquitin ligases.
Author Response
Dear Reviewer,
We greatly appreciate your help in improving our manuscript. Listed below is our point-by-point response.
The only concern is that the authors haven’t mentioned that p53 has multiple isoforms that are expressed in tumors, and their regulation by ubiquitin ligases.
Response: We appreciate this comment and agree that p53 isoforms are a group of important regulators in cancer. At least 12 isoforms have been described with different transcription start sites or generated by alternative splicing. Owing to the deletions of different functional domains, these isoforms may play differential roles in cancer. However, the modes of action of the p53 isoforms are still largely unclear and the paucity of evidence reveals the regulation of the isoforms by E3 ligases except for MDM2. We believe this is an interesting topic increasingly attracting more attentions, and additional biochemical, molecular and cellular studies are desired in the field.